# Effects of Phytase Supplementation to Diets with or without Zinc Addition on Growth Performance and Zinc Utilization of White Pekin Ducks

**DOI:** 10.3390/ani9050280

**Published:** 2019-05-25

**Authors:** Youssef A. Attia, Nicola F. Addeo, Abd Al-Hamid E. Abd Al-Hamid, Fulvia Bovera

**Affiliations:** 1Arid Land Agriculture Department, Faculty of Meteorology, Environment and Arid Land Agriculture, King Abdulaziz University, Jeddah 21589, Saudi Arabia; yaattia@kau.edu.sa; 2Department of Veterinary Medicine and Animal Production, University of Napoli Federico II, 80137 Napoli, Italy; nicolafrancesco.addeo@unina.it; 3Department of Animal and Poultry Production, Faculty of Agriculture, Damanhour University, Damanhour, Egypt; abd_abd_el_hamid_agric@yahoo.com

**Keywords:** ducks, zinc oxide, phytase, growth performance, zinc utilization

## Abstract

**Simple Summary:**

The environment sustainability of farms is extremely important for the future of the world. In this context, the lowering of the pollution from intensive poultry farms is necessary. Due to its low levels and low digestibility in feeds, Zn is often overdosed in poultry feed, and its excess in the excreta can accumulate in the soil, inhibiting the growth of soil microorganism as well as altering their morphology and metabolism, thus reducing the crop yield and quality. Enzymes, such as phytase, can breakdown the linkage of Zn with phytic acid in vegetable feeds, thus increase the Zn availability for animal digestion. In this way, very low supplementation of Zn to the diets can meet the requirement of poultry.

**Abstract:**

The effect of phytase and inorganic Zn supplementation was studied in 180 male White Pekin ducks (WPD) from 1 to 56 days of age. The birds were divided into four groups fed the same basal diet (containing 26 ppm of Zn from raw materials): the control group did not receive Zn supplementation; the second group was supplemented with 30 ppm of Zn oxide; and the third and fourth groups were fed the control and the 30 ppm diets, respectively, both supplemented with 500 U of *E. coli* phytase. Each group contained five replicates of nine ducks. The body weight and feed intake were recorded at 1, 28 and 56 days of age. At 56 days of age, five birds/group were used to measure feed digestibility and five other birds/group were slaughtered. Zn at 30 ppm increased the body weight gain (BWG, *p* < 0.01) and feed intake (*p* < 0.05) and improved the feed conversion (FCR, *p* < 0.05) of the growing ducks. The Zn retention and Zn level in the excreta increased (*p* < 0.01) due to Zn supplementation. The addition of phytase improved BWG (*p* < 0.01) and FCR (*p* < 0.05) of growing ducks. The use of phytase reduced (*p* < 0.01) the level of Zn in duck excreta. Phytase supplementation to the basal diet at 30 ppm seems to be adequate to meet Zn requirements for ducks without further Zn additions.

## 1. Introduction

Zinc (Zn) is an essential trace mineral with several roles in animal metabolism, acting as a functional component of more than 200 enzymes [1,2]. In the NRC guidelines [3], Zn requirements for ducks are not provided; therefore, the dietary requirement of Zn for ducks is based on those for other bird species.

In poultry nutrition, Zn is required for eggshell deposition [4]; inadequate amount of Zn negatively affects the feed intake, growth rate and feed conversion ratio of broilers [5]. In addition, abnormalities in the immune responses, as well as reproduction, skeletal and skin disorders can be tied to the deficiency of Zn in poultry diets [6].

In general, the level of Zn in feedstuffs is low [3] and in vegetable products this element is also poorly available for digestion because its chelation to the phytic acid [7]. Thus, the addition of Zn to poultry diets is a common practice. A comparison between NRC [3] recommendation and modern commercial strains of broilers suggests that industries often use a big safety margin of Zn in feed formulation [5], also considering that amount of Zn up to 2000 mg/kg in poultry diets does not negatively affect the bird performance [8]. However, high amount of Zn in the diets is responsible of the high excretion of this trace element into the environment [9] as fecal Zn content linearly increases with Zn dietary levels [10]. Thus, the European Commission has recently established a maximum limit for the total Zn content, including the supplemental premix, of poultry diets at 100 ppm [11]. Therefore, the knowledge of the specific Zn requirements can reduce its supplementation in poultry diets, without affecting animal health, welfare and productivity [2].

A possible solution could be the use of enzymes associated with vegetables. Phytase is a useful additive that improves the nutritive value of feedstuffs rich in phytic acid and also reduces environmental pollution related to nitrogen, and several metals (Cu, Zn, Fe, and Mg) by improving their availability to the animal and decreasing their excretion into the environment [12,13,14,15,16]. Yu et al. [17] indicated that phytate reduces the Zn absorption in the broiler intestinal tract; therefore, it can be hypothesized that adding the phytase to the diets, the amount of Zn available for poultry digestion can be increased.

The objective of this study was to investigate the effects of phytase supplementation to diets with or without Zn addition on productive performance and physiological traits of growing ducks. The addition of phytase to the diet with Zn oxide aimed to verify if only the supplementation of inorganic Zn is enough to sustain animal performance or if more Zn can provide further improvements.

## 2. Materials and Methods

### 2.1. Experimental Design, Birds, Diets, and Husbandry

All procedures were approved by the Animal and Poultry Production Department, Faculty of Agriculture, Damanhour University (Egypt) that recommends animal rights and welfare.

One hundred eighty one-day-old male White Pekin ducks (*Anas platyrhynchos* domestica, WPD) were homogeneously distributed into four groups fed the same starter and finisher diets (basal diets). The groups were subjected to four dietary treatments as follow: the first group (control) was fed basal diets unsupplemented with Zn oxide or phytase; the second group was fed the basal diets supplemented with 30 ppm of Zn oxide (72% Zn); the third group was fed the basal diets supplemented with 500 U of *E. coli* phytase (*E. coli* 6-phytase, 500 U/kg diet; Danisco Animal Nutrition, England); and the fourth group was fed the basal diets supplemented with 30 ppm of Zn oxide and 500 U of *E. coli* phytase. The basal (starter and finisher) diets were obtained by using a Zn-free trace mineral mixture and contained 26 ppm of Zn from raw materials without Zn supplementation, as measured by Atomic Absorption Spectrometry analysis. The starter diet (fed from 1 to 35 days of age) and the finisher diet (36–56 days of age) were formulated according to NRC [3] recommendations and their ingredients and chemical-nutritional characteristics [18] are reported in Table 1.

Each diet was fed to five replicates consisting of nine male WPD each. Each replicate was housed in floor pens (1.0 m × 2 m) furnished with wood shavings. The brooding temperature was 34, 32, 30 and 28 °C during Weeks 1, 2, 3 and 4, respectively, and thereafter the temperature inside the house was about 27 °C. The light program provided 24 h of light on the first day; and then the lighting was gradually reduced to 10 h/day at 21 days of age. The light was supplied continuously. Water and mash form of feed were offered ad libitum.

### 2.2. Data Collection

The ducks were individually weighed at 1, 28, and 56 days of age in the morning, before offering feed. The remaining, scattered and consumed feed were measured during the periods 1–28, 28–56 and 1–56 days for each replicate; thus, the average feed intake per bird was calculated as the ratio between feed intake and the number of ducks per replicate. The feed conversion ratio (FCR) was calculated as units of feed intake required to produce one unit of gain in live body weight in the periods 1–28, 28–56 and 1–56 days. The mortality rate was recorded along the entire experimental period. At the end of the trial (56 days of age), five birds per treatment were randomly chosen, weighed after being fasted overnight, and slaughtered according to the Islamic guidelines. Feathers were plucked, the inedible parts (head, feet, and inedible viscera) were removed and the remaining (dressed) carcass was weighed. The feathers, liver, spleen, gizzard, heart, pancreas, and abdominal fat were separated and individually weighed. The percentage carcass yield and the percentages of internal organ weights relative to live body weight were calculated. A 50/50 (*w*/*w*) sample of skinless breast and thigh meat was weighed and kept in an electric drying oven at 70 °C until a constant weight was reached. The dried flesh was finely ground through a suitable mixer, passed through a sieve (1 mm^2^), and then carefully mixed and stored in tightly sealed glass containers for subsequent analysis. The physical characteristics of a sample mixture of breast and thigh meats were evaluated. The ability of meat to hold water (WHC) and meat tenderness were measured according to the methods of Volvoinskaia and Kelman [19]. The pH was measured as described by Aitken et al. [20]. The color intensity (optical density) of meat was determined according to the method of Husani et al. [21].

At 56 days of age, five ducks per group were housed in individual cages and used to evaluate the nutrient digestibility of the experimental diets. The birds were housed in individual cages. The methodology involved a four-day adaptation period followed by a three-days excreta collection period. After each day of collection, the excreta samples were dried to come to equilibrium with the atmosphere, weighed, ground and, finally, mixed together and stored in screw-top glass jars until analysis. The proximate chemical composition of the feed and excreta was according to the official methods of Association of Official Analytical Chemists (AOAC) [18].

The Zinc content was determined after ashing of the samples with 10 mL of concentrated sulfuric acid. Three drops of bichloric acid were added and the samples were incubated at room temperature for 2 h. Zinc concentration in the diets, liver, bones, excreta, and plasma were determined by atomic absorption spectroscopy (GBC Avanta Z, GBC Scientific Equipment, Braeside, Australia) using a standard curve. The apparent retention of Zn was calculated by dividing the difference between the amount consumed and that excreted by the amount consumed.

Blood samples were collected from wing vein from five ducks per treatment and placed into heparinized tubes. The plasma was separated by centrifugation at 1500× *g* for 15 min and stored at −18 °C until analysis. The plasma levels of Zn and Cu were determined by atomic absorption spectrometry after processing the samples as previously described.

### 2.3. Statistical Analysis

The data were analyzed using a two-way ANOVA of the General Linear Model (GLM) procedure of SAS [22] in which Zn and phytase supplementations were the main effects. The potential interactions between the effects were also evaluated. A probability of less or equal to 0.05 was considered significant, based on the Student Newman–Keuls Test of mean differences among treatments [22]. The data are reported based on the main effects and significant interactions. The differences among mortality rate were analyzed by chi-square test.

## 3. Results

The grower and the finisher basal diets used in the trial contained 26 ppm of Zn from raw materials (Table 1) as determined by atomic absorption spectrometry. The data on in vivo performance are reported in Table 2.

The mortality rate was not statistically different among the experimental groups. The addition of 30 ppm of Zn to the basal diets increased the body weight gain (*p* < 0.01) and feed intake (*p* < 0.05), and improved the FCR (*p* < 0.05) of ducks considering the entire period of the trial. The supplementation of phytase also improved (*p* < 0.01) BWG and FCR from 1 to 56 days, but the feed intake was not different from the control group. Except for the feed intake, the interaction between the two tested factors was significant: when no Zn was included in the diet, the addition of phytase improved both FCR and BWG; however, when 30 ppm of Zn were added to the basal diet, the addition of phytase did not improve the duck performance.

The addition of Zn to the diets reduced (*p* < 0.01) Zn retention (Table 3) and increased the level of Zn in tibia (*p* <0.01), liver (*p* < 0.05) and excreta (*p* < 0.01).

In addition, the level of Zn and Cu in the plasma increased (*p* < 0.01) due to Zn inclusion in the basal diets. The addition of phytase increased the level of Zn in tibia (*p* < 0.05) and liver (*p* < 0.01) as well as the concentration of Zn and Cu in plasma (*p* < 0.01) but decreased the Zn content in the excreta (*p* < 0.01). The interaction between Zn level and phytase was significant for tibia ash, plasma Zn and plasma Cu. The use of phytase significantly decreased the tibia ash when 30 ppm of Zn oxide were added to the diets, but it did not happen for the Zn-free diet. The addition of phytase to the basal diet increased Zn and Cu concentration by 13.4% and 34.4%, respectively, while the addition of phytase to 30 ppm Zn diets increased the Zn and Cu plasma levels by 9.4% and 29.7%, respectively.

The addition of Zn to the basal diet decreased (*p* < 0.01) the percentage of gizzard but the other carcass traits were unaffected (Table 4). The use of phytase decreased the percentage of liver (*p* < 0.01) and abdominal fat (*p* < 0.01). There was a significant interaction between Zn level and phytase supplementation on gizzard percentage. Results indicate that phytase increased gizzard percentage of ducks fed 30 ppm Zn diet but had no effect when added to the basal diets.

## 4. Discussion

The natural presence of Zn in the diets from the raw materials is not enough alone to adequately sustain the duck growth. In our trial, the addition of 30 ppm of Zn oxide to the basal diets improved the animal performance: the increase of feed intake was responsible for the increased body weight gain, giving a more favorable FCR. Cufadar and Bahtiyarca [23] indicated that 30 ppm of Zn was adequate for growth performance of male WPD. The live weight of the ducks at the end of the trial (56 days) was lower than the data recorded in the literature. However, as reported by Dodu [24], the imported breed of ducks, along the years, was mixed with local populations giving genetic lines differing for some growing characteristics from the original breed. In particular, Dodu [24] indicated that the body weight at 56 days of some Pekin ducks bred in Romania was around 2 kg. In a previous study, Attia et al. [25] found similar body weight for 56-day-old Pekin ducks breed in Egypt.

The supplementation of Zn to the basal diet strongly increased its amount in the excreta, also due to the lowering of the retention rate. In fact, the primary mechanism of trace minerals homeostasis is the modification of the trace minerals absorption and excretion in the gut [25,26,27]. Cao et al. [28] observed that bone and fecal Zn contents were significantly increased when the diets of chickens were supplemented with organic and inorganic sources of Zn. The significant increase of gizzard percentage due to the addition of Zn in the present trial could be justified by the increased feed intake which could play a physical effect on gizzard expansion.

The improved BWG and FCR in ducks supplemented with phytase diets were not due to an increase of feed intake: in fact, the percentage of gizzard was also unchanged between the groups. The positive effect of phytase on growth performance of WPD could be attributed to the increase in the availability of others inorganic and organic nutrients [14,29,30,31]. The positive effect of Zn on BWG was probably due to an improved activity of the copper-requiring metalloenzymes, such as ceruloplasmin, cupro-zinc superoxide dismutase and cytochrome c oxidase, which have a very important role as anti-oxidants in the metabolism [32]. In addition, looking at the interaction effect, the use of the basal diet without Zn supplementation induced lower growth rate than that with an addition of 500 U of phytase or 30 ppm of Zn. The increased growth rate due to phytase or Zn supplementation resulted in an improved FCR. The phytase improved Zn utilization, as evidenced by the increase of Zn in plasma and its decrease in the excreta, but the effect on Zn retention was weak. In the literature, the effect of phytase on Zn availability is contradictory: phytase is reported to increase the availability of dietary Zn [4,23] as well as the bone Zn content in pigs and chicks [33], but to have no significant effect on Zn digestibility and apparent absorption percentage of Zn, Fe, or Cu in chicks [34]. These differences could be ascribed to the differences in the metabolism among the species, the different dietary composition or Zn level in the basal diet. Dietary Zn at 800 ppm negatively affected phytate breakdown by phytase [35] as a result of a conformational change in the phytate moiety, thereby making it less accessible to phytase.

The effect of phytase on plasma Zn content was stronger in WPD fed the basal diet than in those fed diets supplemented with 30 ppm of Zn oxide (13.3% vs. 9.4%). These results are consistent with those reported by Mohana and Nys [36]. In addition, the value for Zn retention found herein agrees with those reported by other authors [10,36]. Similar to the present findings, Jondreville et al. [15] found that 100 FTU of phytase were equal to 1 ppm of Zn, and that the Zn excretion could be reduced by 10% if a corn–soybean diet were supplemented with 500 FTU phytase/kg diet.

In the present study, the phytase significantly increased plasma content of Cu, according to Attia et al. [37,38]. Revy et al. [30] reported a positive effect of phytase on Cu availability due to the effect of phytase on phytate-mineral complex formation. However, Jondreville et al. [15] reported that microbial phytase had a negative effect on the liver Cu content in chicks and pigs, likely because of the negative effect of Zn on Cu availability due to release of Zn by phytase [39].

The lower percentage of the abdominal fat and liver in WPD fed diets with phytase may be attributed to the reapportioning of nutrients towards growth rather than fat accumulation. Similar results were reported by Attia et al. [12,29]. Furthermore, Cufadar and Bahtiyarca [23] reported that increasing dietary phytase at three dietary Zn levels increased the results for all carcass parameters, although the effects were not proportional to the level of dietary phytase; rather, phytase prevented the deleterious effects of dietary Zn on carcass traits. This might explain the positive effect of phytase on the growth and the decrease of the abdominal fat of WPD in the present study. Orban et al. [40], Attia et al. [12,29] and Cufadar and Bahtiyarca [23] found that phytase significantly increased the carcass weight, neck, thigh, back + breast and wings of broilers.

## 5. Conclusions

The natural presence of Zn in raw materials is not enough alone to satisfy the Zn requirements of the growing ducks. The addition of 30 ppm of Zn or 500 U of phytase to the basal diet increased the growth rate and improved the FCR of the ducks. However, the addition of 30 ppm of Zn oxide also increased the level of Zn in the excreta, while the addition of 500 U of phytase had an opposite effect and is more appropriate to reduce the potential risks for environmental pollution.

## Figures and Tables

**Table 1 animals-09-00280-t001:** Ingredients and chemical-nutritional characteristics of the basal diets fed to White Pekin ducks during the starter (1–35 days of age) and the finisher (36–56 days) periods.

Ingredients (kg/ton)	Starter Diets		Finisher Diets	
Yellow corn	564.0	564.0	680.0	680.0
Soybean meal, 44%	383.0	383.0	267.0	267.0
Dicalcium phosphate	20.0	20.0	20.0	20.0
Sunflower oil	15.0	15.0	15.0	15.0
Limestone	10.0	9.58	10.0	9.58
Zinc Oxide	0.0	0.420	0.0	0.420
Salt	3.0	3.0	3.0	3.0
Vit+Min premix *	3.0	3.0	3.0	3.0
DL-methionine	1.0	1.0	1.0	1.0
Antifungal	1.0	1.0	1.0	1.0
Composition (calculated values)				
Metabolizable energy, MJ/kg	12.05	12.05	12.60	12.60
Methionine, g/kg	4.4	4.4	3.9	3.9
Methionine + Cysteine (SSA), g/kg	7.9	7.9	6.9	6.9
Lysine g/kg	11.8	11.8	9.0	9.0
Calcium g/kg	9.5	9.5	9.2	9.2
Available phosphorous g/kg	4.5	4.5	4.3	4.3
Composition (measured values)				
Dry matter, g/kg	894.1	894.1	897.5	897.5
Crude protein, g/kg	212.6	212.6	173.6	173.6
Ether extract, g/kg	42.1	42.1	43.8	43.8
Crude fiber, g/kg	46.1	46.1	46.2	46.2
Ash, g/kg	77.0	77.0	78.4	78.4
Nitrogen free extract, g/kg	593.3	593.3	633.9	633.9
Zn, ppm	26.0	26.0	26.0	26.0

* Vit + Min Premix provides the following (per kg of diet): Vitamin A, 1800 mg retinol; Vitamin E, 6.67 mg d-alpha-tocopherol; menadione, 2.5 mg; Vit D3, 50 mcg cholecalciferol; riboflavin, 2.5 mg; Ca pantothenate, 10 mg; nicotinic acid, 12 mg; choline chloride, 300 mg; vitamin B12, 4 mcg; vitamin B6, 5 mg; thiamine, 3 mg; folic acid, 0.50 mg; biotin 0.2 mg; Mn, 80 mg; Fe, 40 mg; Cu, 4 mg; Se, 0.10 mg.

**Table 2 animals-09-00280-t002:** Effect of zinc supplementation, with and without phytase addition, on body weight gain, feed intake and feed conversion ratio of ducks *.

Treatment Group	BWG (g/Bird/Period)			Feed Intake (g/Bird/Period)			FCR (g/Bird/Period)			Dead (*n*)
	1–28 day	29–56 day	1–56 day	1–28 day	29–56 day	1–56 day	1–28 day	29–56 day	1–56 day	
Zn addition										
0 ppm	1010 ^b^	916 ^b^	1926 ^b^	1931 ^b^	4182 ^b^	6114 ^b^	1.91	4.59	3.18 ^a^	3
30 ppm	1079 ^a^	965 ^a^	2044 ^a^	1991 ^a^	4307 ^a^	6297 ^a^	1.85	4.46	3.08 ^b^	1
Phytase inclusion										
0 U/kg diet	1032	910 ^b^	1942 ^b^	1961	4247	6209	1.91	4.67^a^	3.20 ^a^	2
500 U/kg diet	1056	970 ^a^	2026 ^a^	1960	4242	6202	1.86	4.38^b^	3.06 ^b^	2
Interaction between Zn and phytase										
0 ppm Zn + 0 U phytase	974 ^d^	855 ^b^	1830 ^b^	1937	4189	6126	1.99 ^a^	4.90 ^a^	3.35 ^a^	2
0 ppm Zn + 500 U phytase	1045 ^c^	976 ^a^	2021 ^a^	1924	4175	6100	1.84 ^b^	4.28 ^b^	3.02 ^b^	1
30 ppm Zn + 0 U phytase	1089 ^a^	966 ^a^	2054 ^a^	1985	4305	6291	1.82 ^b^	4.46 ^b^	3.06 ^b^	0
30 ppm Zn +500 U phytase	1068 ^b^	965 ^a^	2032 ^a^	1995	4308	6303	1.87 ^b^	4.47 ^b^	3.10 ^a,b^	1
*p* value										
Zn addition	0.0003	0.0023	0.0004	0.0295	0.0364	0.0336	0.1071	0.0711	0.0471	NS
Phytase inclusion	0.1191	0.0005	0.0007	0.9529	0.9221	0.9323	0.2344	0.0021	0.0077	NS
Interaction	0.0069	0.004	0.0002	0.6578	0.8849	0.8089	0.0289	0.0022	0.0012	NS
RMSE	33.15	30.6	45.17	55.23	121.6	176.5	0.094	0.149	0.105	0.94

* *n* = 180 bird as 45 bird per treatment group for body weight gain and *n* = 20 replicates as five replicates per each treatment for feed intake and feed conversion ratio. RMSE, Root mean square error. ^a–d^ means with different superscripts in the same column in similar treatment group are significantly different; NS, not significant.

**Table 3 animals-09-00280-t003:** Effect of zinc supplementation, with and without phytase addition, on Zn retention and tissue and excrement concentrations, and plasma Zn and Cu concentrations in White Pekin ducks *.

	Zn Retention (%)	Tibia Ash (%)	Tibia Zn (ppm)	Liver Zn (ppm)	Excrement Zn (ppm)	Plasma Zn (mg/100 mL)	Plasma Cu (mg/100 mL)
Zn addition							
0 ppm	37.67 ^a^	44.65	161.3 ^b^	61.3 ^b^	72.15 ^b^	160.1 ^b^	6.61 ^b^
30 ppm	35.17 ^b^	45.13	165.4 ^a^	63.4 ^a^	160.0 ^a^	189.3 ^a^	7.22 ^a^
Phytase inclusion							
0 U/kg diet	35.90	45.15	162.8 ^b^	61.0 ^b^	117.6 ^a^	164.9 ^b^	5.93 ^b^
500 U/kg diet	36.96	44.63	164.0 ^a^	63.7^a^	114.3 ^b^	184.0 ^a^	8.28^a^
Interaction between Zn and phytase							
0 ppm Zn + 0 U phytase	36.64	44.46 ^b^	160.6	60.7	74.06	149.2 ^d^	5.23^d^
0 ppm Zn + 500 U phytase	38.70	44.91 ^a^	161.9	61.8	70.06	169.0 ^c^	7.97 ^b^
30 ppm Zn + 0 U phytase	35.12	45.91 ^a^	165.0	61.4	161.2	180.7 ^b^	6.63 ^c^
30 ppm Zn +500 U phytase	35.22	44.36 ^b^	165.7	65.5	158.4	197.6 ^a^	8.60 ^a^
*p* value							
Zn addition	0.0008	0.2795	0.0001	0.0367	0.0001	0.0001	0.0001
Phytase concentration	0.0918	0.2429	0.0458	0.0126	0.0001	0.0001	0.0001
Interaction	0.1232	0.0302	0.5068	0.1378	0.4466	0.0006	0.0001
RMSE	1.346	0.958	1.072	2.12	1.491	0.736	0.164

* *n* = 20 samples as five sample per each treatment. RMSE, Root mean square error. ^a–d^ means with different superscripts in the same column in similar treatment groups are significantly different; NS, not significant.

**Table 4 animals-09-00280-t004:** Effect of zinc supplementation, with and without phytase addition, on the percentage weights of the dressed carcass, carcass parts, inner organs, and abdominal fat in White Pekin ducks *.

	Dressing (%)	Front Part (%)	Hind Part (%)	Pancreas (%)	Spleen (%)	Liver (%)	Gizzard (%)	Heart (%)	Abdominal Fat (%)
Zn addition									
0 ppm	65.4	41.4	24	0.333	0.061	2.25	3.79 ^a^	0.66	0.618
30 ppm	65.5	41.7	23.8	0.388	0.050	2.17	3.05 ^b^	0.616	0.641
Phytase concentration									
0 U/kg diet	66.6	42.3	24.2	0.352	0.055	2.39 ^a^	3.42	0.638	0.790 ^a^
500 U/kg diet	64.4	40.7	23.7	0.369	0.056	2.03 ^b^	3.41	0.614	0.472 ^b^
Interaction between Zn and phytase									
0 ppm Zn + 0 U phytase	67.5	43.1	24.4	0.373	0.058	2.31	4.02a	0.693	0.820
0 ppm Zn + 500 U phytase	63.4	39.7	23.7	0.291	0.064	2.18	3.56 ^a,b^	0.626	0.413
30 ppm Zn + 0 U phytase	65.6	41.6	23.9	0.332	0.051	2.46	2.83 ^c^	0.622	0.748
30 ppm Zn + 500 U phytase	65.4	41.7	23.7	0.446	0.048	1.88	3.26 ^b^	0.602	0.530
*p* value									
Zn addition	0.9343	0.1353	0.7878	0.3446	0.075	0.5699	0.0001	0.5213	0.6869
Phytase concentration	0.0941	0.0729	0.4483	0.7761	0.8531	0.0123	0.9465	0.2596	0.0001
Interaction	0.1293	0.0602	0.7413	0.1188	0.4321	0.093	0.0288	0.5684	0.9421
RMSE	2.712	1.881	1.548	0.132	0.013	0.28	0.413	0.255	0.119

* *n* = 20 samples as five sample per each treatment; RMSE, Root mean square error. ^a–d^ means with different superscripts in the same column in similar treatment groups are significantly different; NS, not significant.

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
