# Peer review of "Effects of Phytase Supplementation to Diets with or without Zinc Addition on Growth Performance and Zinc Utilization of White Pekin Ducks"

_animals, 2019, doi:10.3390/ani9050280_

Round 1
Reviewer 1 Report
General Comments
The authors observed white Pekin duck performance when fed diets supplemented with zinc and phytase enzyme. The experimental design is sound and the data collected was sufficient to make necessary conclusions on the study. The manuscript is well written with the exception of the simple summary. The materials and methods are also presented clearly although there is concern on the number of times data was collected throughout the study. Was data collected three times only or weekly? The objective of this study needs clarification…what would be the purpose of evaluating phytase supplementation in diets supplemented with Zinc oxide? Phytase supplementation in diets with high phytin complexes of phosphorus that would also complicate Zn bioavailability is understandable. The results and discussion are also presented clearly. There are a few concerns as presented in the specific comments below:
Specific Comments:
P1L18 …Zn in the excreta can accumulate in the environment…..state the consequences of such accumulation. This entire section of the manuscript should be re-written.
P1L20 …in the why, also very low…..not sure what this sentence means…
P1L25 …present all 4 experimental diets in Table 1; did all 4 diets contain the 26 ppm Zn from raw material?
P2L57 ..associated with vegetables…….
P2L73 …submitted to the following dietary…..replace “submitted” to “subjected”
P2L73-76…rewrite for clarity of dietary treatments……
P3L99-98…feed intake was measured weekly? How about body weights?
P4L120-121…. Rewrite this sentence.
P4L133…A probability of less or equal to 0.05 ….
P4L137 …grower and finisher…….
P46L140….mortality rate was unaffected by dietary treatments…….explain
P4L147 …addition of Zn reduced Zn retention…is the reduction in total amount of ZN retained in tissues? The representation of % Zn retention in this study as presented in Table 3 is not accurate as relates to the objective of this study.
Author Response
The authors observed white Pekin duck performance when fed diets supplemented with zinc and phytase enzyme. The experimental design is sound and the data collected was sufficient to make necessary conclusions on the study. The manuscript is well written with the exception of the simple summary. The materials and methods are also presented clearly although there is concern on the number of times data was collected throughout the study.
Thank you very much for your comments.
Was data collected three times only or weekly? The data were collected 3 times and this has been reported in the text (lines 97-98 and line 100).
The objective of this study needs clarification…what would be the purpose of evaluating phytase supplementation in diets supplemented with Zinc oxide? The aim has been clarified (lines 65-68)
Phytase supplementation in diets with high phytin complexes of phosphorus that would also complicate Zn bioavailability is understandable. The results and discussion are also presented clearly.
Thank you very much
There are a few concerns as presented in the specific comments below:
Specific Comments:
P1L18 …Zn in the excreta can accumulate in the environment…..state the consequences of such accumulation. This entire section of the manuscript should be re-written.
The consequences have been indicated (lines 18-19). The section has been partly re-written
P1L20 …in the why, also very low…..not sure what this sentence means…
there was a mistake (corrected)
P1L25 …present all 4 experimental diets in Table 1; did all 4 diets contain the 26 ppm Zn from raw material? Yes analyses revealed that
P2L57 ..associated with vegetables…….done
P2L73 …submitted to the following dietary…..replace “submitted” to “subjected” done
P2L73-76…rewrite for clarity of dietary treatments……the period has been rewritten (lines 77-81)
P3L99-98…feed intake was measured weekly? How about body weights? These points have been clarified (lines 97 -100)
P4L120-121…. Rewrite this sentence. Rewrite (line 126)
P4L133…A probability of less or equal to 0.05 ….corrected
P4L137 …grower and finisher…….corrected
P46L140….mortality rate was unaffected by dietary treatments…….explain explained (line 147)
P4L147 …addition of Zn reduced Zn retention…is the reduction in total amount of Zn retained in tissues? The reduction is in the total amount of Zn as explained at lines 130-131.
The representation of % Zn retention in this study as presented in Table 3 is not accurate as relates to the objective of this study. A period has been added to clarify this point (lines 221-222).
Reviewer 2 Report
This manuscript suffers from low originality and poor novelty. This information and analytical tools are used are not fashionable.
Author Response
This manuscript suffers from low originality and poor novelty. This information and analytical tools are used are not fashionable.
We are sorry for this comment, in our opinion this topic is very actual and need to be further studied in poultry
Reviewer 3 Report
The work simple, but it was well planned and described in a manuscript that is very easy to read and follow. In my opinion, it is recommended for publication.
Author Response
The work simple, but it was well planned and described in a manuscript that is very easy to read and follow. In my opinion, it is recommended for publication.
Thank you very much for your comment
Round 2
Reviewer 1 Report
None
Author Response
Dear reviewer,
thank you very much. The English has been improved.
Reviewer 2 Report
The authors revised their work substantially.
The current version satisfies the high scientific level of the publication.
Author Response
dear reviewer, thank you very much. The English has been corrected.